# STEGO NETWORKS:
# INFORMATION HIDING ON DEEP NEURAL NETWORKS

## ABSTRACT

*The best way of keeping a secret is to pretend there is not one.* In this spirit, a class of techniques called steganography aims to hide secret messages on various media leaving as little detectable trace as possible. This paper considers neural networks as novel steganographic cover media, which we call stego networks, that can be used to hide one's secret messages. Although there have been numerous attempts to hide information in the output of neural networks, techniques for hiding information in the neural network parameters themselves have not been actively studied in the literature. The widespread use of deep learning models in various cloud computing platforms and millions of mobile devices as of today implies the importance of safety issues regarding stego networks among deep learning researchers and practitioners. In response, this paper presents the advantages of stego networks over other types of stego media in terms of security and capacity. We provide observations that the fraction bits of some typical network parameters in a floating-point representation tend to follow uniform distributions and explain how it can help a secret sender to encrypt messages that are indistinguishable from the original content. We demonstrate that network parameters can embed a large amount of secret information. Even the most significant fraction bits can be used for hiding secrets without inducing noticeable performance degradation while making it significantly hard to remove secrets by perturbing insignificant bits. Finally, we discuss possible use cases of stego networks and methods to detect or remove secrets from stego networks.

## 1 INTRODUCTION

As much as it goes without saying knowledge is power, inventing methods for keeping and selectively conveying secret messages has been a crucial mission throughout the history of humanity. Among various methods to protect secrets, an effective approach called *steganography* makes it difficult to detect the very existence of the secrets in an object looking innocuous. The object containing the secrets is called a *stego medium* in the context of steganography.

Starting with the case of hiding a secret message in the form of engraved tattoos on hidden parts of a human body in an ancient greek period, numerous methods (e.g., using invisible inks, writing tiny-sized letters) were employed to transmit information without leaving detectable footprints (Kahn, 1996). Most recently, digital steganography, which embeds secret messages in digital images or audio files, has been actively developed. Traditional steganography is typically used in communication between two individuals, but steganography in digital media enables its brand-new usage by conveying secrets in a multitude of devices and unknowingly influencing their behavior when accompanied with a small decoding code. The secrets in this scenario are often called *stegomalware* (Nagaraja et al., 2011; Suarez-Tangil et al., 2014).

Meanwhile, deep neural networks (DNNs) have shown remarkable success in various areas over the years, and are now beginning to be applied to industry and the consumer sector as well as to the academic. DNNs have been deployed in a variety of computing systems ranging from large-scale cloud computing systems to millions of mobile devices (Howard et al., 2017). More and more mobile devices are running application programs that include deep learning models with numerous camera filter and speech recognition applications being good examples. Furthermore, building upon existing large pre-trained models, such as ResNet (He et al., 2016), BERT (Devlin et al., 2019), and

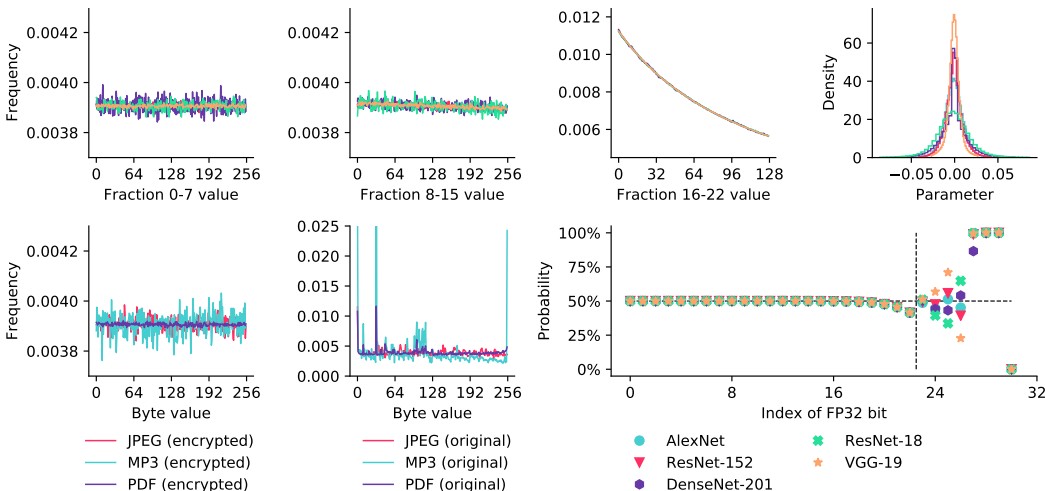

Figure 1: Fraction bit distributions of image classification model parameters and distributions of evaluated bytes of various file formats (JPEG, MP3, and PDF). We visualized the distributions of fraction bits of parameters into three different plots. Three subsequences of 23 fraction bits are evaluated at their decimal values (**top row, first three**). The distributions of the entire parameters of each model is also presented (**top row, last one**). Next, we plotted byte distributions of the original and the encrypted files (**bottom row, first two**). The plotted values are averaged over 100 files for each file format. Finally, the probabilities of each bit in floating-point representation to be one is visualized (**bottom row, last one**). An interesting observation is that the fraction bits of five architectures follow nearly identical distributions, but the shape of parameter distributions are noticeably different. Exponent bits of floating-point numbers mainly responsible for the discrepancy between parameter distributions.

GPT-3 (Brown et al., 2020), rather than training complete neural networks from scratch has become a trend in deep learning research. Accordingly, various files containing neural network parameters are uploaded on the source code repositories, and FTP servers, and they are frequently exchanged among individuals and organizations.

The fraction bits, which are sometimes called the significand or the mantissa, of a floating-point number follows a distribution close to uniform distribution as shown in Figure 1. A secret sender can easily embed encrypted messages, which also typically follows a uniform distribution, without causing much suspicion from static analysis tools. Also, as the sizes of pre-trained neural network parameters are often large being more than hundreds of megabytes in size, neural network parameters are suitable media to exchange a nontrivial amount of secrets.

In this paper, we analyze the distribution of fraction bits of typical neural network parameters. Fraction bits, which contain the least significant information of a floating-point number, can conveniently be used to embed secret messages. We experimented with a special kind of weight perturbation, which simulate general cases of hiding secrets in network parameters and explored several methods to inject arbitrary data into neural network parameters without noticeable performance degradation. We empirically showed that steganography in the least significant fraction bits is readily applicable and even steganography in the most significant fraction bits is also possible. Our main contributions are as follows:

- We demonstrate suitability of neural network parameters as a novel steganographic medium.

- We propose novel approaches to effectively embed secrets in neural network parameters.

- We give comprehensive analysis of stego networks in terms of security and capacity, and discuss the advantages of stego networks over the conventional stego media.

## 2 BACKGROUND

This section covers materials to facilitate understanding of this paper, which includes the fundamentals of steganography and related work.

### 2.1 STEGANOGRAPHY

Steganography is a technique to conceal information from unauthorized persons or eavesdroppers. It is a sub-field of information hiding. The goal of steganography is to hide the presence of the secret by embedding it in a camouflage medium, which is called a *cover medium*. There exist three aspects in steganography that decide its effectiveness: *security*, *capacity*, and *robustness* (Provos & Honeyman, 2003; Cox et al., 2007). Among them, security is often considered the most important factor. Security means that the presence of the secret must not be revealed under any situation. The next one, capacity, means the amount of information that the steganographic system can carry. Usually the risk of being detected increases as capacity increases. Robustness means the durability from external noise or interruption to eliminate the embedded information. In the context of stego networks, robustness can be degraded if re-training or fine-tuning is applied.

Among various cover media, we mainly compare stego networks with images because it is one of the most actively studied media of secret information and image files are widely-used file formats among deep learning researchers and practitioners. Usually, each pixel of a three-channel RGB image is represented by a total of 24 bits, eight bits per channel. A trivial approach to hide a secret image in a cover image is to replace the least significant bits in the pixel values of the cover image with the most significant bits of the secret image. However, this approach is vulnerable because the secret image can be seen in the stego image due to humans' highly sensitive visual perception. The process that reveals the existence of steganography by visual characteristic is called *visual attack*. Also, digital images follow a certain distribution of pixel values that can be easily distinguished.

*Steganalysis* refers to the process of detecting the steganography by capturing the statistical anomaly of data. Several techniques have been proposed such as a subtractive pixel adjacency matrix (SPAM) (Pevny et al., 2010) to neutralize the steganographic systems. Currently, well-known steganography algorithms include HUGO, WOW, and S-UNIWARD (Pevnỳ et al., 2010; Holub & Fridrich, 2012; Holub et al., 2014).

### 2.2 RELATED WORK

There have been numerous approaches in the literature that applied deep neural network to steganalysis. The objective of some of these studies was to construct neural networks, which can determine whether hidden contents is embedded in an input image (Xu et al., 2016; Yedroudj et al., 2018). On the other hand, Baluja (2017) and Zhu et al. (2018) proposed methods to generate stego images. Our work is clearly different from these studies in that our work considers the neural network itself as a cover medium of steganography.

To the best of our knowledge, there has not been a principled study of neural networks as cover media of steganography so far. One work relevant to of ours is *LSB embedding attack* proposed by Song et al. (2017). It the work, a scenario is suggested where an adversary intercepts training data with compromised training algorithm which hides the data in the LSFBs of the trained neural network parameters. The data embedding method can be seen as a special form of LSFB steganography which we explain in Section 3. However, Song et al. (2017) does not include discussion of neural network parameters in terms of steganographic security, capacity, and robustness, which are the three main aspects of steganography. Therefore, it can hardly be seen as study of properteis of neural networks as steganographic cover media.

Our work is closely related with research on sensitivity and precision of neural network parameters. Cheney et al. (2017) conducted a sensitivity analysis on the convolutional neural networks (CNNs) and reported that AlexNet (Krizhevsky et al., 2012) layers close to the input showed relatively high sensitivity to weight perturbation than those layers close to the output. Weight quantization (McKinstry et al., 2018; Wang et al., 2018; Zafrir et al., 2019; Jung et al., 2019) is also an actively studied research area for energy and storage efficiencies. Studies in this field showed that neural networks

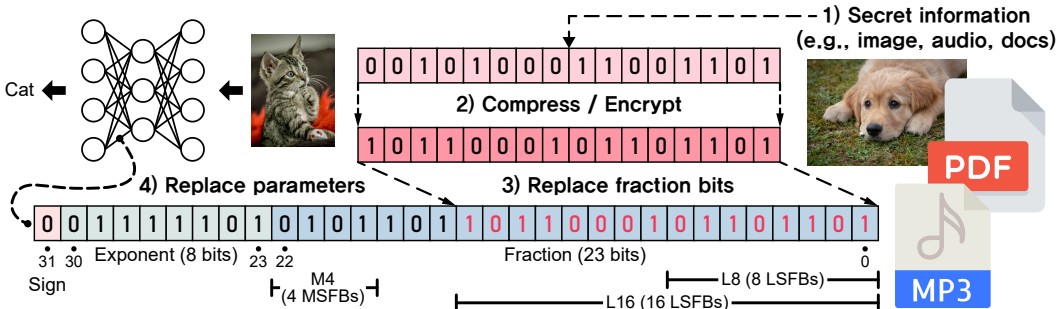

Figure 2: Hiding information in neural network parameters. Before embedding, secrets should be compressed and encrypted to provide additional protection. A hiding procedure is to simply replace fraction bits of the parameter with the encoded message. In this paper, we convey information either in 1) the most significant 4 fraction bits (denoted by **M4**), or in 2) the least significant 8 or 16 fraction bits (denoted by **L8** and **L16**, respectively) of fraction bits.

can preserve their performance although a half or less number of bits are used to represent parameters.

In terms of DNN security, weight poisoning (Gu et al., 2017) and adversarial attack (Goodfellow et al., 2015) also have been frequently studied. Both methods aim to manipulate the output of neural networks, but the difference between the two methods is that weight poisoning perturbs neural network parameters while adversarial attack perturbs the input of the network. In the case of adversarial attack, its counter-measures called adversarial defense (Metzen et al., 2017) and their counter-counter-measures have also been continuously studied (Ghiasi et al., 2020).

Another closely related field to our work is watermarking. Both watermarking and steganography belong to the area of information hiding. While the top priority of steganography is security, watermarking focuses on robustness, in order to prevent attempts to removing information stored in cover media. Watermarking on neural networks can be used to integrity check and copyright protection of the network. Adi et al. (2018) prepared a trigger dataset, which is completely unrelated to the original task, and train or fine-tune a model to yield 100% accuracy on the trigger set. Then, the auditor can determine whether the model is watermarked or not by checking the accuracy of the trigger set. Similarly, Le Merrer et al. (2020) adjusted the decision boundary of the model for specific inputs. Instead of examining the output, there exists a method that probe the activation values from the networks for a private input (Darvish Rouhani et al., 2019). Lastly, Uchida et al. (2017) added a regularization loss on the network parameters to embed watermark. The watermark is a form of a binary vector, which can be recovered by multiplying an embedding matrix with the flattened parameters. In the case of watermarking, it usually leaves the noticeable trace on the network parameters, thus it cannot transmit information confidentially.

## 3 STEGO NETWORKS

This section presents the analysis on the sample distributions of fraction bits of typical neural network parameters. Afterwards, we propose two of our novel approaches to embed secret messages in neural networks.

### 3.1 FRACTION BITS OF NEURAL NETWORK PARAMETERS

Neural network parameters are usually in floating-point types. A floating-point number consists of three parts for sign, exponent and fraction bits[1]. For example, a single-precision (FP32) format has one sign bit, eight exponent bits, and 23 fraction bits. Since FP32 is one of the most widely used formats in deep learning research, we use the format throughout this paper. The least significant fraction bit has the lowest index, i.e., zero. Since embedding messages in sign or exponent bits

---

[1]Details of floating-point number are described in *IEEE Standard for Binary Floating-Point Arithmetic*.

can induce a relatively significant perturbation to the original value, we only consider the case of embedding bits in fraction bits.

The fraction bit distributions of the parameters of a few models commonly used in computer vision are provided in Figure 1. We divide the 23 bits of fraction bits into three parts indexed as 0-7, 8-15, and 16-22, to ease the visualization. We represent a sequence of bits of each part simply by their corresponding decimal value. For example, a bit sequence 10000010 with the right-most bit denoting the bit of the lowest index is represented as 130 in Figure 1.

When it comes to randomness of fraction bits, we particularly considered two issues, which are uniformity of a random variable and independent and identically distributed (i.i.d.) property of a sequence of random variables. We can denote the part of FP32 fraction bits indexed 0-7 as $X_i$ with $|X_i| = 256$, and $i$ is the index of the corresponding FP32 number in the entire parameters. Then one may think $X_i$ is a uniform random variable if she observes each outcome happens with almost the same frequency. On the other hand, if one often observes the same outcomes consecutively, she may think the sequence of $X_i$ is not i.i.d. If $X_i$ is uniform and a sequence of $X_i$ are i.i.d., then the sequence has the maximum entropy.

Note that a message embedder usually encrypt secret messages to provide additional protection. Also, encrypting messages makes it difficult for a steganalysis system to detect the existence of the message since much of the traces disappears. Encryption algorithms such as AES and RSA are often involved before message embedding. In the rest of this section, we assume message embedders always encrypt secret messages.

Uniformity of a random variable is directly related with the capacity of stego networks and an secret embedder may need to sacrifice the bit rate if the bit distribution of a target medium is not uniform. Figure 1 shows distributions of the three fraction parts of five popular models in computer vision. The two least significant parts show nearly uniform distributions across all the models. The most significant seven bits seemingly followed non-uniform distributions. In this case, a secret sender may want to add dummy bits in the original secret message to avoid a discrepancy between the fraction bit distribution of the original neural network and the resulting stego network to ensure security.

Another important property relevant to steganographic security is the i.i.d. property. This property is what various steganalysis algorithms are based on to detect anomaly in stego media. For example, SPAM algorithm (Pevny et al., 2010) utilizes the fact that pixels of natural images have high spatial correlation such that the colors of neighboring pixels have similar values. Therefore, if the usual bits of a cover medium are purely i.i.d., it automatically wards off a large number of sophisticated steganalysis systems. Note that establishing theoretical independence between the significands of parameters is out of scope in our work. Instead, we mainly provide empirical results that has practical implications in the experiment section. From the experimental results, we suspect that significands of stego networks exhibit considerable low dependency between them compared to the storages of other popular steganographic media.

## 3.2 HIDING SECRETS IN THE LEAST SIGNIFICANT FRACTION BITS

We first propose embedding in neural network parameters by putting secrets in the least significant fraction bits (LSFBs). We recommend to measure the sensitivity of target neural network parameters before embedding messages by perturbing different amounts of fraction bits of the parameters to decide the precise number of bits that can be used for steganography. As can be observed in Section 4, a large portion of LSFBs in popular image classification models can be perturbed without noticeably affecting the validation accuracy.

An obvious drawback of this approach is that secrets can be easily threatened by an active warden, an agent who aims to eliminate the chance of the existence of secrets, by simply perturbing the LSFBs. Also, applying this method can be tricky if targeted network parameters have significands with low-enough precision such that messages cannot be injected without inducing significant performance degradation.

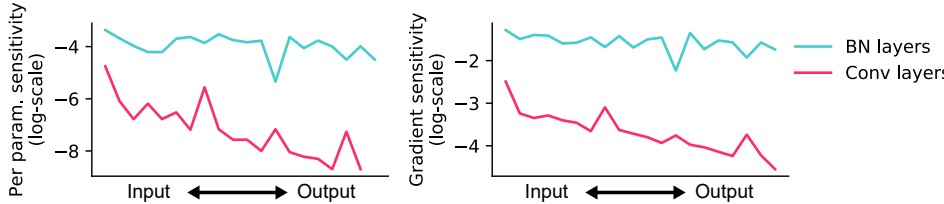

Figure 3: Validation accuracy sensitivity and gradient sensitivity of the layers of ResNet-18. The more a layer is to the left, the closer it is to input. Due to significant scale differences between BN and convolutional layers, y-axes are represented in base-10 log scale. Overall, a layer closer to the output has a larger sensitivity in either metric.

### 3.3 HIDING SECRETS IN THE MOST SIGNIFICANT FRACTION BITS

We also propose an opposite-directional approach from the above which is to hide secrets in the most significant fraction bits (MSFBs) of a floating-point number. This method perturbs the original parameter much more than when hiding secrets in the LSFBs. For example, when perturbing the most significant one bit the value of the perturbed parameter can become $3/2$ or $2/3$ times of the original value.

Since changing the MSFBs can result in significant performance degradation of a model we suggest several remedies. First, since each layer of a neural network has different sensitivities to random perturbations we recommend one to conduct sensitivity analysis to choose proper parameters. Also, fine-tuning can be followed to compensate performance degradation after embedding the secret. In Section 4, we partitioned the parameters of a model into two where the former is used to hide secrets into one part while the latter is fine-tuned.

Another due consideration is the distribution of the most significant fraction bits. If they do not follow a uniform distribution, a message sender may need to add dummy bits as described earlier. However, adding non-uniform dummy bits violates i.i.d. property of the encrypted message and may increase the risk of being detected, assuming the original distribution also has the i.i.d. property.

Although this approach has more challenges than the other approach, it also has a clear advantage that the message cannot be simply removed by perturbing insignificant bits since it can significantly degrade the model performance.

## 4 EXPERIMENTS

This section provides experimental results and demonstrate the usefulness of DNNs as cover media. First, we conducted a preliminary experiment by measuring sensitivity (Cheney et al., 2017; Widrow & Lehr, 1990) of each layer of the ResNet-18 architecture. Next, we applied the secret embedding methods proposed in the previous section for larger models to simulate real-world scenarios. All the values reported in the table are the averaged ones across ten different runs.

Throughout the experiments, we perturbed network parameters by injecting independent random bits which has the equal probability to be $0$ or $1$. Although it is just a simple setting, it can actually represent general cases since it is common to encrypt messages before embedding using encoding algorithms generating uniformly random bit patterns.

### 4.1 SENSITIVITY ANALYSIS

The sensitivity of each layer of ResNet-18 was measured after perturbing all the fraction bits of the corresponding layer's parameters. We defined and used two different sensitivity metrics to discover parameters causing minimum effects after a message is embedded. The first one measures the decrease in validation accuracy due to the perturbation to a specific layer and divides that value by the number of the parameters of that layer. The latter one computes the parameter gradients with respect to the loss function over a large-sized input. The gradients are multiplied by the magnitudes of the parameters to reflect the fact that parameters are perturbed by a relatively large amount when the

| Layer type | | Ratio | Avg. per param. sensitivity | Avg. gradient sensitivity |
|---|---|---|---|---|
| Conv | Weight | 95.53 | $1.139 \times 10^{-6}$ | $4.005 \times 10^{-4}$ |
| FC | Weight | 4.38 | $7.000 \times 10^{-8}$ | $3.299 \times 10^{-4}$ |
| | Bias | 0.01 | $4.000 \times 10^{-8}$ | $6.961 \times 10^{-5}$ |
| BN | Weight | 0.04 | $\mathbf{1.495 \times 10^{-4}}$ | $\mathbf{2.883 \times 10^{-2}}$ |
| | Bias | 0.04 | $8.408 \times 10^{-5}$ | $9.381 \times 10^{-3}$ |

Table 1: Summary of sensitivity analysis on ResNet-18 architecture. We measured the sensitivity of all layers in the model. Then, we categorized and averaged the result by the type of layers. Batch normalization layers, which accounts for 0.08% of the total number of parameters are particularly more sensitive to the output.

| Model | | Accuracy | | | | |
|---|---|---|---|---|---|---|
| | | Original | **L8** | **L16** | **M4** | **M4** (tuned) |
| AlexNet | Top-1 | 56.522 | **56.524** (.001) | 56.501 (.018) | 56.388 (.040) | 56.388 (.072) |
| | Top-5 | 79.066 | 79.066 (.002) | **79.090** (.012) | 79.047 (.024) | 79.047 (.040) |
| VGG-19 | Top-1 | 72.376 | 72.376 (.001) | 72.368 (.013) | **72.379** (.026) | 72.379 (.046) |
| | Top-5 | **90.876** | **90.876** (.000) | **90.876** (.009) | 90.871 (.018) | 90.871 (.033) |
| ResNet-152 | Top-1 | 78.312 | 78.311 (.001) | 78.309 (.018) | 76.836 (.074) | **78.368** (.061) |
| | Top-5 | **94.046** | **94.046** (.000) | 94.032 (.008) | 93.277 (.068) | 94.034 (.033) |
| DenseNet-201 | Top-1 | 76.894 | **76.898** (.002) | 76.896 (.021) | 70.660 (.226) | 76.796 (.075) |
| | Top-5 | 93.370 | 93.371 (.001) | **93.384** (.016) | 90.406 (.139) | 93.375 (.040) |

Table 2: Accuracy of the original and the perturbed image classification models (Krizhevsky et al., 2012; Simonyan & Zisserman, 2015; He et al., 2016; Huang et al., 2017). We measured accuracies of various model architectures with four perturbing strategies on ImageNet validation dataset.

values of the parameters are are larger than those of other parameters in our setting. The rectified gradients are averaged in each layer. Finally, we changed the signs of both metrics so that they always have positive values. ILSVRC2012 (Russakovsky et al., 2015), also known as ImageNet, was used to compute validation accuracy and average gradients. The average sensitivity values of the same layer type are reported in Table 1. We found that the sensitivity of batch normalization (BN) layer is 10 to 100 times larger than other layers in both metrics. Since the number of BN parameters only account for less than 1% of the entire parameters, which indicates low capacity, we excluded BN parameters in later experiments. In addition, we plotted the trend of the sensitivity along the layers of a network. Figure 3 shows that, in general, the layers closer to the output have lower sensitivity than the layers close to the input. A similar tendency is also reported in Cheney et al. (2017).

## 4.2 LSFB AND MSFB PERTURBATIONS

We compared four perturbation approaches by assessing the validation accuracy of the resulting models. We took four pre-trained ImageNet classification models (AlexNet, VGG-19, ResNet-152, and DenseNet-201) and perturbed their parameters. The result described in Table 2 shows that the least significant 8 fraction bits (**L8**) and the least significant 16 fraction bits (**L16**) perturbation do not significantly affect the model accuracy in all cases. On the other hand, ResNet-152 and DenseNet-201 showed remarkable degradation in accuracy in the case of the most significant 4 bits (**M4**) perturbation (we perturbed only 50% of the parameters for **M4** experiment). We fine-tuned the models (**M4**) for one epoch, and ResNet-152 and DenseNet-201 largely recovered the original accuracy.

| Layer type | | # of layers | | Max. Z-score | | Avg. Z-score |
| --- | --- | --- | --- | --- | --- | --- |
| | | | Frac. 0-7 | Frac. 8-15 | Frac. 16-22 | Image |
| Conv | Weight | 20 | 3.877 | 4.528 | 3.943 | |
| FC | Weight | 1 | 1.053 | 0.653 | 2.083 | 626.747 |
| | Bias | 1 | 0.744 | 1.645 | 0.327 | |
| BN | Weight | 20 | 2.109 | 1.992 | 2.478 | |
| | Bias | 20 | 2.147 | 2.004 | 2.048 | |

Table 3: Result of runs tests on ResNet-18 parameters and Imagenet validation images. A high Z-score can be interpreted as an evidence that the corresponding sequence of the variables are not i.i.d.

## 4.3 PARAMETER DISTRIBUTION

We observed the parameter distribution after perturbation and fine-tuning (see Appendix A). In the case of **M4**, it clearly left the trace in the bit distribution while **L8** kept the resulting bit distribution nearly indistinguishable from the original one and **L16** left a few footprints. Those traces stayed even after fine-tuning, thus appending dummy bits may be needed to avoid detection especially when applying **M4**.

Also, to check the i.i.d property of one set of fraction bits to another, a runs test (Bradley, 1968), a statistical procedure that determines whether a sequence of observations are mutually independent, was conducted on the parameters of ResNet-18 (He et al., 2016) in a simple setting. Fraction bits of a floating-point number were converted into the usual three decimal values as before and only the mutual independence between one type of decimal values were tested as shown in Table 3. We also measured the Z-scores of the runs test in ImageNet validation data for comparison. The Z-score of the parameters of a layer was separately calculated and only the maximum Z-scores among the same type of layers were reported. In deciding positiveness in the testing, we used the mean value of a sequence as a reference. The resulting Z-scores for the parameters are still too high in the usual statistical sense to accept the null hypothesis that the sequence variables are independent. However, there was a significance difference between Z-scores from the neural network parameters and the image data. We suspect this relatively little dependency among parameter fraction bits makes it fundamentally harder to detect anomaly from neural network parameters than from image data.

## 4.4 DEEP LEARNING STEGANALYSIS TO DETECT STEGO NETWORKS

We trained deep learning models to distinguish stego network parameters from innocuous neural network parameters. Although there exist a few benchmark datasets for image steganography such as BOSS (Bas et al., 2011), no suitable dataset for the task of detecting stego networks seem to exist at the moment. Hence, we used the parameters of the fully connected layer of four pre-trained ImageNet classification models (AlexNet, VGG-19, ResNet-152, and DenseNet-201) as our training and test datasets and split them into the ratio of 8 to 2. In particular, we further divided the datasets into two types, one consisting only of the most significant fraction bits and the other consisting of the least significant fraction bits. Windows of size $32 \times 32$ cropped from the parameter matrix of size $1000 \times 512$ were fed to the models we trained. Both the original parameters and parameters with perturbation in

| Model | Method | Best accuracy | |
| --- | --- | --- | --- |
| | | **L16** | **M4** |
| AlexNet | MLP | 51.063 | 50.485 |
| | Yedroudj-Net | 50.917 | 54.780 |
| VGG-19 | MLP | 50.686 | 50.539 |
| | Yedroudj-Net | 52.229 | 54.776 |
| ResNet-152 | MLP | 50.748 | 50.710 |
| | Yedroudj-Net | 51.646 | 55.449 |
| DenseNet-201 | MLP | 50.434 | 50.271 |
| | Yedroudj-Net | 51.421 | 54.506 |

Table 4: Performance of two deep learning steganalysis models on stego networks classification. Note that dummy bits were used in **M4**.

the entire fraction bits were included as negative and positive (stego) input data, respectively. More details about the experiment can be found at Appendix A.

We experimented with a custom multilayer perceptron (MLP) and a popular image steganalysis model named Yedroudj-Net (Yedroudj et al., 2018). As can be seen in Table 4, both models generated output close to random predictions. The result indicates that the significands of the fully connected layers do not follow a probability distribution that can be easily distinguished from a random bit sequence. Please note that this experiment is intended as a preliminary one and, therefore, cannot be interpreted as a definite evidence for security of stego networks. We expect more specialized methods for stego networks are developed in the future works.

## 5 DISCUSSION & CONCLUSION

In this paper, we explored stego networks, a novel application of neural networks in steganography. First, we analyzed parameters of various image classification models and discovered that the fraction bits of the parameters tend to follow a uniform distribution. We also observed that the sequence of the fraction bits exhibit a degree of independence in the sense of the Z-score of a runs test. These two fundamental characteristics are what, we suspect, give stego networks outstanding advantages over the other conventional stego media.

In the experiment section, we applied our suggested methods to widely used deep learning models to simulate embedding secrets in neural networks. We confirmed that there was not significant performance degradation even when one half of the total bits of a neural network were perturbed. In addition, we suggested best practices in embedding messages in MSFBs. Sensitivity analysis was particularly helpful by exhibiting relatively high sensitivity of batch normalization layers and layers close to input. With additional help of fine-tuning, we showed that a neural network can large maintain its original performance after perturbations in MSFBs. Finally, we reported our experiment where an existing deep learning based steganalysis technique could not be effectively applied to distinguish compromised neural network parameters from normal neural network parameters.

A potential application of steganography is to embed malware into innocuous-looking media turning them into stegomalware (Nagaraja et al., 2011; Suarez-Tangil et al., 2014). A typical stegomalware consists of an actual stegomalware and its decoding program of a tiny size. It is generally more involved to identify stego malware by static analysis than identifying usual obfuscated malware. Currently, there exist a tremendous amount of pre-trained models and open-source code on the Internet. According to Verdi et al. (2019) and Zimmermann et al. (2019), a large number of publicly available source code repositories contain vulnerabilities, but numerous software developers still employ the codes without doubts. In the worst case, pre-trained models can be a prevalent hacking vulnerability enjoying the obscurity of stego networks at the moment. Also, more and more mobile devices are running application programs that include deep learning models with numerous camera filter and speech recognition applications being good examples. This implies that the general public can be a target of malicious stego networks. We attached a practical demonstration of a malicious stego network to Supplementary Material for proof-of-concept.

Finally, we provide a few useful counter-measures to either detect or eliminate secret messages. First, perturbing the entirety or a part of the parameters of a model by a small amount can effective eliminate hidden messages in the LSFBs without significant performance degradation. In the case of hiding messages in MSFBs, one may conduct a static analysis and and detect the existence of abnormal information by observing frequencies of certain bit patterns or dependencies among a series of significands.

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

# A   APPENDIX

This section provides additional materials, which are not described in the main manuscript.

## A.1   DEEP LEARNING BASED STEGANALYSIS

We used a 3-layer multilayer perceptron (MLP) with 1024 and 128 hidden features and two dropout layers of the dropout rate of 0.3. We trained both our model and Yedroudj-Net for 30 epochs using stochastic gradient descent with the learning rate of 0.01 and the weight decay of 0.0001.

When processing the input data, we made a few peculiar design choices. Given a floating-point number we discarded the sign and exponent bits and only kept the fraction bits. The fraction bits were again divided into two parts as in the previous experiments, the four most significant 4 bits (**M4**) and the least significant 16 bits (**L16**). **L16** was treated as two 8-bit sequences rather than one 16-bit sequence. Finally, we evaluated them as integers and composed two types of samples, one consisting of evaluated MSFBs of the size $32 \times 32 \times 1$ and the other consisting of two evaluated less significant bit sequences of the size $32 \times 32 \times 2$. The values of the elements of MSFB samples range from 0 to 15, and those of LSFB sample from 0 to 255.

When simulating secret embedding (actually, simply perturbing bits) in **M4**, we also added dummy bits to make the embedded bits indistinguishable from normal MSFBs. We first randomly generate a finite number of dummy intervals. They are used in rotation to access a next parameter. When dummy intervals are one, three and two, for example, parameters at indices $1, 4, 6, 7, 10, 12, ...$ will store dummy bits. In this experiment, we generated 1024 random dummy intervals and used the same intervals for both train and test datasets. We added randomly chosen dummy bits into M4 parts of a stego network such that the eveluted M4 integers of a stego network has the same integer frequencies as that of a corresponding cover network. Note that this is a unnatural setting since, in practice, a warden does not have information about the dummy intervals a message embedder used. In a more realistic and complex setting, the number of dummy intervals can be much less while retaining the equivalent performance.

## A.2 PARAMETER DISTRIBUTION

The two plots below illustrate the change of the marginal probability distributions of each bit of a parameter before and after fine-tuning. The most significant four fraction bits (**M4**) are mostly likely to be detected by steganalysis due to their larger difference from the original networks marginal probabilities. Fine-tuning does seem to help close the discrepancy between the original and stego network marginal bit distributions although the difference is still significant enough to be detectable by a warden.

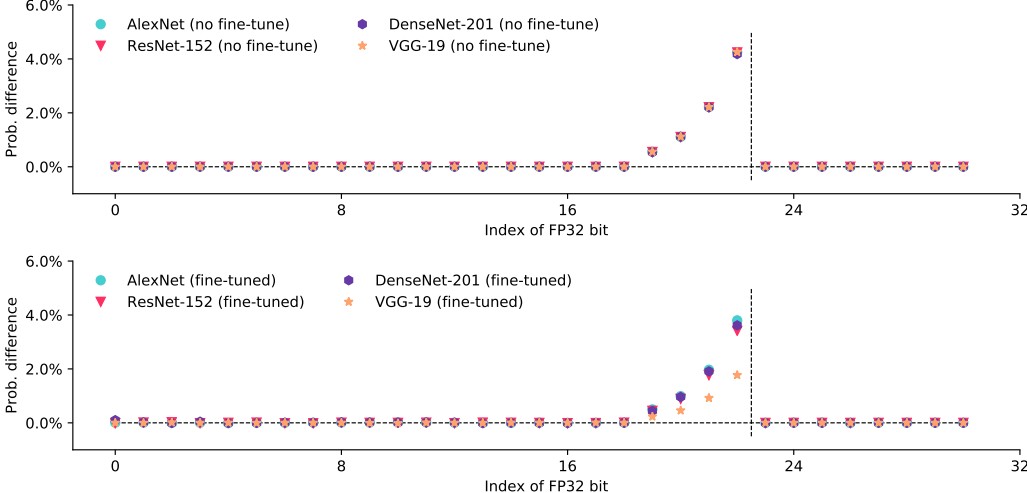

Figure 4: Change in the probabilities of each bit of a floating-point number to be one. The plot in the top shows the difference of probabilities between a perturbed model distribution and the original model. The plot in the top shows the difference of probabilities between a perturbed and fine-tuned model and the original model.

