# OpenReview forum: "Stego Networks: Information Hiding on Deep Neural Networks"
_ICLR.cc/2021/Conference — Reject_

### Official Review · AnonReviewer3 · 2020-10-25
**Stego Networks: interesting but unclear motivation**

**Rating:** 3
**Confidence:** 3

**Review:**

This paper investigates if neural network parameters, trained for a standard task such as image recognition, can be used as a cover medium in steganography. The authors contend that neural networks are a good choice of cover medium, mainly because the less important fractional bits of parameters follow a uniform distribution. This is empirically demonstrated in Figure 1 where the probability of least significant bits taking values 0 or 1 are 50% for VGG, ResNet, DenseNet (I assume these are ImageNet models judging from later experiments). The authors experiment with replacing either the least or most significant bits from the fractional part of parameter values. The motivation behind replacing the most significant bits is that it will be more difficult for an attacker to remove secret information; bits of lower significance could be removed or replaced with little effect on test accuracy, while removing more significant fractional bits will have a larger impact on accuracy. The downside of this approach is that replacing more important fractional bits with secrets will also have a larger impact on accuracy, but the authors show fine-tuning can somewhat alleviate this drawback. The authors then show simple MLPs trained to distinguish between standard and stego networks are only slightly better than a random coin flip. Overall, I thought the paper was generally well-written and contains some interesting ideas, but these positives are also accompanied by an unclear motivation, lack of positioning with respect to related work, and poor experimental evaluation.

My primary concern is the practical motivation for this idea. In steganography, it is normally assumed that both the cover and steg image/audio/network cannot both be revealed, otherwise a trivial comparison will reveal secret information is present thus breaking security of the scheme. Does this mean a new model needs to be trained each time two parties want to communicate? If so, this seems to represent a serious limitation, since training these models requires a non-trivial amount of effort, in comparison to say, generating an image or audio file as the cover. Furthermore, a problematic scaling law then appears, where larger models need to be trained (on presumably more complex tasks to recover some plausible deniability e.g. it would be suspicious to see a ResNet-50 trained on MNIST), to hide larger messages. These larger models are much more expensive to train, and again generating high-resolution images is a much cheaper option. Sharing large resolution images will generally consume less memory than very large neural networks. I wonder how the authors view the practicality of this work? It would have been great to compare this scheme with some standard steganography schemes on images or audio across desideratum such as bandwidth, robustness etc.

The authors may not be aware, but there has been work on information hiding in previous work. Song et al. (2017) [1] also investigate how information can be imperceptibly embedded and recovered from neural networks. In addition to experiments on embedding secrets in fractional bits, they have experiments in black-box settings. Could the authors comment on relationship between this work and Song et al. (2017) [1]? As far as I can tell, they are quite similar ideas.

I was a little disappointed that the authors didn't include any practical demonstrations, such as successful recovery of secrets as shown in Song et al. (2017) [1]? Steganographic scheme are evaluated by security, capacity and robustness, as the authors comment in Section 2.1, but most of the experiments in section 4 concentrate on capacity. How does the scheme stand-up against robustness attacks such as down-stream task fine-tuning, parameter pruning etc.? I was also confused by the experiment set-up in Section 4.4. For a fair experiment of distinguishability, I had expected a large number of cover and stego models to be trained (for some notion of confidence), and then use a steganalysis tool to distinguish between the two. I'm not sure training a steganalysis model on a single stego network allows for a fair interpretation of performance.


[1] Song, Congzheng, Thomas Ristenpart, and Vitaly Shmatikov. "Machine learning models that remember too much." Proceedings of the 2017 ACM SIGSAC Conference on Computer and Communications Security. 2017.

---

> ### Author Response · Authors · 2020-11-22
> **Response to Reviewer 3 (1/3)**
>
> ***I assume these are ImageNet models judging from later experiments***
>
> In Figure 1, all the models were pre-trained using ImageNet. We will clarify this in the revised version.
>
> ---
>
> ***My primary concern is the practical motivation for this idea....***
>
> We admit it is a valid concern that communications using stego networks require a new model for every message communication. That is, stego networks may not be the best option when one has to frequently exchange a small amount of secret information. However, we would like to point out that frequent and light communications are not the only scenario where steganography can be effectively used. For those who want to deliver a large-sized secret in one file, stego networks can be the only possible option since it is much more difficult for other stego media to deliver messages of a few hundred megabytes when faced with the current state-of-the-art detecting methods.
>
> The success of a particular stego medium largely depends on how much the file format is actually consumed among people. A main motivation for our study was the realization that a considerable number of neural network models are frequently uploaded and exchanged on the internet as mentioned in the introduction. For example, public repositories such as PyTorch Hub (https://pytorch.org/hub/) and Transformers by Hugging Face (https://huggingface.co/transformers/) contains numerous models from different sources and a large number of deep learning researchers and practitioners use the repositories' pre-trained models. This large-scale many-to-many communication is clearly contrasted with sequential peer-to-peer communication. There also exist other public repositories maintained by individuals.
>
> According to Verdi et al. (2019) [1], a large number of publicly available source code repositories contain vulnerabilities, but numerous software developers still employ the codes without doubts. Also, Zimmermann et al. (2019) [2] pointed out that numerous codes usually rely on a small number of centralized open-source libraries. "left-pad" incident discussed in Zimmermann et al. showed that the entire internet system can be halted even when only one tiny block of code has a problem. In this circumstance, adversaries can enjoy the obscurity of stego networks in these days by fabricating ML-related libraries and distributing contaminated pre-trained models (stego-malware in neural networks).
>
> Likewise, when combined with malware, stego networks can pose a serious threat to a large audience. Famous historical examples of stegomalware include Visbot Malware Found on 6,691 Magento Online Stores. Nearly 6,700 online stores running on top of the Magento platform are infected with the Visbot malware that hides on web servers and steals credit card information. We were substantially worried about a similar usage of stego networks and wanted to call for widespread awareness. Millions of mobile applications such as FaceApp contain neural networks in them. It means not only deep learning researchers but also the general public can easily be a target of malicious attacks through stego networks. As is mentioned in our general response, a demo of stegomalware was added for the proof of concept.
>
> Furthermore, unlike stego images, which have been analyzed for decades, stego networks are a newly proposed concept, and their properties and countermeasures are largely underexplored. In image steganography, one has to make sure message embedding does not result in perceivable changes in the image. Anomalies in neural network parameters cannot be easily perceived by human eyes, which makes it significantly easier to embed messages.
>
> We generally acknowledge that the current manuscript did not provide sufficient motivation, so we will improve Sections 1 and 5 by giving additional concrete examples and detailed analysis based on this discussion. Also please note again that we do not suggest stego networks as a replacement for all the other cover media, but just as a new medium for different use cases.
>
> [1] Verdi, Morteza, et al. "An empirical study of c++ vulnerabilities in crowd-sourced code examples." IEEE Transactions on Software Engineering (2020).
>
> [2] Zimmermann, Markus, et al. "Small world with high risks: A study of security threats in the npm ecosystem." 28th USENIX Security Symposium (2019).

---

> > ### Author Response · Authors · 2020-11-22
> > **Response to Reviewer 3 (2/3)**
> >
> > ***The authors may not be aware, but there has been work on information hiding in previous work. Song et al. (2017) [1] also investigate how information can be imperceptibly embedded and recovered from neural networks. In addition to experiments on embedding secrets in fractional bits, they have experiments in black-box settings. ould the authors comment on relationship between this work and Song et al. (2017) [1]? As far as I can tell, they are quite similar ideas.***
> >
> > We thank the reviewer for pointing out the related work (Song et al, 2017). After reading the work, we found that part of Song et al. is relevant to our work although motivated by a quite different goal.
> >
> > The main concern of Song et al. is to prevent untrusted training codes that can steal data without authorization. An attacker can steal training data from a victim when the victim trains a model with training code customized by the attacker. The proposed attack methods are categorized into black-box and white-box. Compared to our work, Song et al.'s discussion of neural networks as secret carriers is clearly limited since it only assumes neural network training data as message material.
> >
> > Among various methods suggested by Song et al., "LSB encoding attack", which is one of the white-box attack methods, can be seen as an example of "hiding secrets in the least significant fraction bit (LSFB)". In other words, a special form of LSFB steganography was proposed before our work. We will make sure to mention this in the revised version.
> >
> > Please note that our primary interest was to demonstrate the suitability of neural network parameters as a steganographic medium. In other words, our work focuses on revealing characteristics of neural network parameters rather than providing sophisticated message embedding algorithms for a special use case. By studying the behavior of neural network parameters under various perturbations, we explained the properties and advantages of stego networks. Song et al. did not include a systematic study of neural network parameters in terms of security, capacity, and robustness which are the three main aspects of steganography. In fact, a single occurrence of the word ‘steganography’ was found in Song et al., and it can hardly be seen as a comprehensive steganographic analysis on neural network parameters.
> >
> > ---
> >
> > ***I was a little disappointed that the authors didn't include any practical demonstrations, such as successful recovery of secrets as shown in Song et al. (2017) [1]?***
> >
> > Actually, the recovery experiments in Song et al. were about "correlated value encoding attack" and "sign encoding attack", which are lossy encoding methods. Since the two methods do not guarantee original data to be intact after encoding, they require recovery experiments. In contrast, "LSB encoding attack" by Song et al. is lossless and thus a recovery experiment is not needed. Therefore, no recovery experiment was conducted for "LSB encoding attack" in their paper.
> >
> > In our work, we only consider lossless encoding and decoding methods. Therefore, we do not think that no recovery experiment is required for our methods, either. However, we decided that it is still a good idea to include a practical demo to interest readers after reading R3's comment. The revised version will include a practical demo.

---

> > > ### Author Response · Authors · 2020-11-22
> > > **Response to Reviewer 3 (3/3)**
> > >
> > > ***Steganographic scheme are evaluated by security, capacity and robustness, as the authors comment in Section 2.1, but most of the experiments in section 4 concentrate on capacity. How does the scheme stand-up against robustness attacks such as down-stream task fine-tuning, parameter pruning etc.?***
> > >
> > > Actually, more than one experiment is related to the security of steganography. We suggested in Section 4.3 that the relatively little dependency among parameter fraction bits makes it difficult to detect anomaly in neural network parameters compared to image data. Also, Section 4.3 is fully devoted to security. Therefore, we do not agree most of our experiments in Section 4 is only about capacity.
> > >
> > > Regarding robustness, it is clear that LSFB steganography is fragile to LSFB perturbation as we mentioned in Section 3.2. On the other hand, the most significant fraction bit (MSFB) steganography is robust to bit perturbations by a warden (or, by R3’s wording, an attacker) as far as the warden wants to keep the performance of the model. However, we do not think MSFB is robust to fine-tuning since it is easy to imagine a few parameters would get much larger gradients than normal. In case of pruning, we think MSFB is obviously vulnerable to it. Please also note that these kinds of vulnerability can be significantly alleviated by using an error-resilient coding scheme. We will augment the discussion about robustness in the revised manuscript.
> > >
> > > ---
> > >
> > > ***I was also confused by the experiment set-up in Section 4.4. For a fair experiment of distinguishability, I had expected a large number of cover and stego models to be trained (for some notion of confidence), and then use a steganalysis tool to distinguish between the two. I'm not sure training a steganalysis model on a single stego network allows for a fair interpretation of performance.***
> > >
> > > We agree that the current experiment is not comprehensive enough to make a definitive conclusion. We will provide additional experiments for the four models in Table 2 in the revised version. However, we believe steganalysis for neural networks cannot be completed in one paper and should continuously be investigated by future works. We will denote the limitation of this experiment in the revised manuscript.

---

### Official Review · AnonReviewer1 · 2020-10-29
**An approach to use the parameters of a neural network to hide information**

**Rating:** 7
**Confidence:** 3

**Review:**

This paper proposed a method to hide information in the parameters of neural network models. To avoid significant perturbation, the paper only considers embed the information in the fraction bits of the parameters. The paper considers hiding the information in either the least significant bits of the most significant bits. Hiding in the least significant bits is harder to be detected but the message can also be easily removed without much degradation in the model performance. On the other hand, information  hiding in the most significant fraction bits will be very hard to remove without model performance degradation but also harder to embed the message for the same reason. Sensitivity analysis to select least sensitive parameters to use and fine-tuning after embedding to recover the model performance can be two remedies.

Overall the paper is clearly written and the proposed method is well supported by the experiments. Some questions / comments below:
- How many files in each format are used to generate the plots in figure 1? Is it statistically significant?
- Sec 3.3 "In the experiment section, we partitioned the parameters of a model into two where the former is used to hide secrets into one part while the latter is fine-tuned." It's not clear to me what the criteria is for partitioning the parameters. Is it purely based on sensitivity analysis?
- Sec 4. "All the values reported in the table are the averaged ones across ten different runs.". Can the standard deviation be added to the results as well?
- Sec 4.1 "Since the number of BN parameters only account for less than 1% of the entire parameters, which indicates low capacity, we excluded BN parameters in later experiments." Does it mean all models used do not have BN layers?
- Re. Table 2: It's interesting to see some models are not sensitive to even MSFB perturbations. Any insight on the specific reason?

---

> ### Author Response · Authors · 2020-11-22
> **Response to Reviewer 1**
>
> ***How many files in each format are used to generate the plots in figure 1? Is it statistically significant?***
>
> In  Figure 1, we concluded that it is infeasible to give a truly objective statistical result for each file format. Therefore, we meant this figure only as an inspiring example. However, we still think that the figure helps readers understand the differences between the file formats.
>
> We used only a few files for each format to create the figure in the original manuscript. In the revised version, we use all time top 100 MP3 files on Free Music Archive (https://freemusicarchive.org/), 100 JPEG images from ImageNet dataset (which are available in the demo), and 100 PDF documents which have been submitted to ICLR 2021 (on OpenReview).
>
> ---
>
> ***Sec 3.3 "In the experiment section, we partitioned the parameters of a model into two where the former is used to hide secrets into one part while the latter is fine-tuned." It's not clear to me what the criteria is for partitioning the parameters. Is it purely based on sensitivity analysis?***
>
> Yes, it is purely based on the sensitivity analysis. We will clarify this in the revised version.
>
> ---
>
> ***Sec 4. "All the values reported in the table are the averaged ones across ten different runs.". Can the standard deviation be added to the results as well?***
>
> The standard deviations for the results range from 0.001 (%) to 0.226 (%).
> We will add the standard deviations to the revised manuscript.
>
> ---
>
> ***Sec 4.1 "Since the number of BN parameters only account for less than 1% of the entire parameters, which indicates low capacity, we excluded BN parameters in later experiments." Does it mean all models used do not have BN layers?***
>
> What we actually meant is that we did not perturb BN parameters in all the experiments except for the sensitivity analysis. All models did use BN layers if the original models had ones. We will make the sentence clearer in the revised version.
>
> ---
>
> ***Re. Table 2: It's interesting to see some models are not sensitive to even MSFB perturbations. Any insight on the specific reason?***
>
> We have an idea for this, and an experiment is being conducted to confirm the idea. We will let the reviewer know the result when the experiment is done.

---

> > ### Author Response · Authors · 2020-11-25
> > **Additional comment for Reviewer 1**
> >
> > ***Re. Table 2: It's interesting to see some models are not sensitive to even MSFB perturbations. Any insight on the specific reason?***
> >
> > Table 2 shows that AlexNet and VGG-19 are not much sensitive to M4 perturbation as opposed to ResNet-152 and DenseNet-201. One conspicuous difference between the two groups is the existence of batch normalization layers (BNs). We compared VGG-19 models with and without BNs to see how the existence of BNs affects performance after M4 perturbation. The top-1 validation accuracy of VGG-19 with BN decreased by 1.29% point. This is contrasted with VGG-19 without BNs, since it showed no performance degradation at all after M4 perturbation. Therefore, we believe that the existence of BNs is mainly responsible for the performance degradation in ResNet-152 and DenseNet-201, too.

---

### Official Review · AnonReviewer4 · 2020-11-02
**Very interesting, novel direction. Experiments could be more thorough.**

**Rating:** 7
**Confidence:** 4

**Review:**

This paper highlights and studies the interesting possibility of hiding information within neural network weights, which is a form of steganography. The sensitivity of different neural network layers to perturbations is evaluated, and based on this a technique for hiding information is proposed and demonstrated. It is shown that it is possible to hide information in the weights of a number of standard baseline neural networks without being easily detectable.

Overall this is an interesting paper, which highlighted a possibility which I was not aware of (although I do have a little familiarity with steganography). My only significant criticism is that I would really like to see a more thorough exploration and discussion of the information hiding capacity of neural networks. In particular it would be great if the authors could explore the relationship between the quantity of information that can be hidden and the size/number of parameters. As far as I can see, the sizes of the weights are only mentioned in passing once (on the first page of the paper), where they are said to be 'in the hundreds of megabytes' (an understatement these days), and the paper does not say how much information they were able to hide in weights. As a bare minimum I would ask that the authors state how much information they were able to hide (as in, how many megabytes), as well as the actual sizes of the weights for the neural networks used (rather than the vague 'hundreds of megabytes'). A more thorough exploration of the relationship between neural net size and stego capacity would be even better. Does it follow the 'square root law' typical in other settings (see e.g. http://www.cs.ox.ac.uk/andrew.ker/docs/ADK71B.pdf)? Perhaps this more thorough exploration can be deferred to future work.

Apart from this I have a number of more minor criticisms and suggested improvements:
 - Multiple issues with Figure 1.
   - In the first sentence of the caption: 'Fraction bit distributions', you should probably replace 'distributions' with 'histograms'.
   - The y-axis of at least one of the histogram plots should be labelled (I guess with 'frequency') so that it's clear what the numerical values represent.
   - Instead of 'top left three' you should be more specific and say 'top row, first three'.
 - Generally there were a lot of grammatical errors and spelling mistakes, and you should spend more time carefully proof-reading the paper. A (probably not exhaustive) list of examples:
   - The first sentence of the second paragraph on page 2 beginning 'Since the fraction bits' is not a valid sentence.
   - 'Neural' is misspelled in the first line of the last paragraph of page 3.
   - The title of Section 2 should be 'Background', not 'Backgrounds'
   - In first sentence of Section 4, what does 'their' refer to? Probably should replace 'their usefulness' with 'the usefulness of DNNs'
   - About two thirds of the way through the first paragraph in Section 4.4, in the sentence beginning 'In particular, ...'; 'fraction bits an...' that 'an' should be an 'and'. Also in 'we made further divided them into two types', delete 'made'.
   - The third sentence of the second paragraph of Section 5, beginning 'In particular, our experiment...' is not a valid sentence.
 - Don't use double parentheses for citations as you do on the last line of page 1. You can use semicolons to separate the citations from the other text inside the parentheses.
 - In the third paragraph of page 3, you say 'A trivial approach to hide... replace the least significant bits in the pixel values of the **secret** image...' I'm pretty sure you mean cover image there, not secret image.
 - In the second paragraph of page 4, could you explain the difference between watermarking and steganography?
 - In the fifth paragraph on page 5 you say '...a load of...', this is far too colloquial for an academic paper.
 - The investigation represented by Table 1 could have been more detailed. For example you could have tested (as others have done previously) whether outputs are more sensitive to weights closer to the input/output of the NN. You could also note that it's pretty obvious that a NN would be more sensitive to batchnorm parameters than others since these affect the scale and position of all of the outputs of a layer.
 - The way you structure the experiments section, with two short introductory paragraphs followed by subsections, could be streamlined. Merge the informative parts of the intro paragraphs into the subsections and then delete the intro paragraphs.

---

> ### Author Response · Authors · 2020-11-22
> **Response to Reviewer 4 (1/2)**
>
> ***In particular it would be great if the authors could explore the relationship between the quantity of information that can be hidden and the size/number of parameters. As far as I can see, the sizes of the weights are only mentioned in passing once (on the first page of the paper), where they are said to be 'in the hundreds of megabytes' (an understatement these days), and the paper does not say how much information they were able to hide in weights. As a bare minimum I would ask that the authors state how much information they were able to hide (as in, how many megabytes), as well as the actual sizes of the weights for the neural networks used (rather than the vague 'hundreds of megabytes').***
>
> We conducted experiments on AlexNet (234 megabytes), VGG-19 (549 megabytes), ResNet-18 (45 megabytes), ResNet-152 (231 megabytes), and DenseNet-201 (78 megabytes). As we mentioned in the manuscript, we did not embed messages in the batch normalization (BN) and bias parameters, but they account for only a small portion of the entire parameters (see Table 1).
>
> When using L16 (embedding in 16 least significant fraction bits), one can embed secret information whose size is nearly 50% of the model size. The capacities of the aforementioned models become 117 megabytes, 274 megabytes, 22 megabytes, 115 megabytes, and 39 megabytes, respectively, when L16 is used. When L8 is used, the capacity becomes the halves.
>
> On the other hand, M4 (embedding in the four most significant fraction bits) is more complicated than the above cases because the message embedder needs to consider (1) fine-tuning and (2) dummy bits to make a stego network indistinguishable from a network without secrets. In order to fine-tune the model, the message embedder should separate model parameters into two sets: one for embedding secret information and the other for fine-tuning. In our experiments (as reported in Table 2), the ratio between the two groups was 1:1 (see Section 4.2). Therefore, 6.25% of the entire parameter bits were used to hide information. The percentage can change if the message embedder adjusts the ratio.
>
> We will clarify these concrete numbers in the revised manuscript.
>
> ---
>
> ***A more thorough exploration of the relationship between neural net size and stego capacity would be even better. Does it follow the 'square root law' typical in other settings (see e.g. http://www.cs.ox.ac.uk/andrew.ker/docs/ADK71B.pdf)? Perhaps this more thorough exploration can be deferred to future work.***
>
> We appreciate the reviewer for providing a good discussion point. In practice, at least at the moment, the application of stego networks is not hindered by the law since there exist no known effective detecting methods for stego networks yet. In steganography experiments, capacity can only be defined by considering a particular security level. Security does not have an absolute definition, either, since it is determined by an eavesdropper's ability. Therefore it is not possible to measure the tradeoff between capacity and security in stego networks.
>
> So far, most of the previous theoretical analysis relied on the assumption that cover media consists of i.i.d random variables. As shown in Table 3 in the manuscript, neural network parameters tend to act as an i.i.d. random variable than image data. Hence, the assumption is naturally applicable to stego networks, which in turn makes the theoretical analysis practically useful. We conjecture that the fraction bits of stego networks also follow the square-root law in the end, but further studies should follow to investigate this issue thoroughly.
>
> Meanwhile, we would like to share interesting discussion points for later studies with the reviewers, as follows. First, the typical example known to follow the square root law is "batch steganography", which is an approach to conceal information by splitting it into several cover media. We think that stego networks can be considered as batch steganography because neural networks consist of multiple fully-connected layers or convolutional layers with multiple channels. Another is the relationship between the performance of neural networks (e.g., validation accuracy or loss) and capacity. Although the performance can be treated as a property of security, it should be treated carefully because neural network parameters have different functionality from other steganographic cover media.

---

> > ### Author Response · Authors · 2020-11-22
> > **Response to Reviewer 4 (2/2)**
> >
> > ***Minor issues (typo, figure, and etc.)***
> >
> > ***The investigation represented by Table 1 could have been more detailed. For example you could have tested (as others have done previously) whether outputs are more sensitive to weights closer to the input/output of the NN. You could also note that it's pretty obvious that a NN would be more sensitive to batchnorm parameters than others since these affect the scale and position of all of the outputs of a layer.***
> >
> > We really appreciate the reviewer for taking time to give insightful and helpful suggestions. We will provide the additional sensitivity analysis results with their comprehensive analysis in the revised version. Also, all issues such as typos will be reflected in the updated manuscript.

---

### Author Response · Authors · 2020-11-22
**General response**

We thank the reviewers for providing constructive comments to improve the manuscript.

Also, we would like to inform the reviewers that we have uploaded a demo to show a practical example of stego networks. There are three source code files ('data.py', 'densenet.py', and 'mal_demo.py'), one model parameter file ('mal_model.pth'), and several images (in 'imgs' directory) for measuring model accuracy. We encourage the reviewers to execute 'mal_demo.py' to see what it does (it is not harmful). Also, the reviewers can check out 'densenet.py' to understand how malicious code snippets (as known as malware) can be hidden in a typical neural network definition.

The demo shows that detecting a malicious code in stego media is more difficult than detecting one in a text file. For example, every system call (e.g., modifying files, managing devices, and controlling processes), which is usually regarded as potentially dangerous, is hidden in the stego network.

We also attach a link to a video, https://figshare.com/s/637f058ee34491242637, for a quick view. The reviewers can check the video to see how the demo works without having to actually download the demo.

---

### Author Response · Authors · 2020-11-24
**Summary of changes in the revised manuscript**

Dear reviewers,

We have updated the manuscript based on the reviews and below is the list of changes. Please note that minor changes (e.g. typos) are omitted.


#### Section 1 (Introduction)

1. We added y-axis labels of fraction bit distribution plots in Figure 1.

2. We clarified notation for indicating sub-plots, e.g., from “top left three” to “top row, first three” in Figure 1.

3. Byte distribution sub-plots that show the original and the encrypted files were re-drawn with averaged values. We used 100 files for each file format in Figure 1.

#### Section 2 (Background)

1. In Subsection 2.1, a paragraph about stegomalware was deleted (this part is moved to Section 5).

2. We added a paragraph to introduce Song et al. in Subsection 2.2.

3. In Subsection 2.2, we supplemented an explanation of the difference between steganography and watermarking.

#### Section 4 (Experiments)

1. We added Figure 3 to show that layers closer to output have lower sensitivity than layers close to input. This result is also described in Subsection 4.1 and Section 5.

2. We provided standard deviations of model accuracies in Table 2.

3. In Subsection 4.4, we conducted additional experiments on four image classification models (AlexNet, VGG-19, ResNet-152, and DenseNet-201) about two embedding methods (L16 and M4). In the case of M4, we utilized dummy bits. Table 4 and Appendix A.1 were also updated to reflect the result.

Section 5 (Discussion and Conclusion)

1. We moved the paragraph about stegomalware from Section 2. Also, we supplemented thoughtful discussion of the worst case scenario of stegomalware.

---

### Decision · Program_Chairs · 2021-01-07
**Final Decision**

**Decision:**

Reject

**Comment:**

This work presents an original analysis of using the weights of a neural network as a medium on which to hide information. Although the paper offers a novel perspective, its motivation and applicability remain unclear. As reviewer 3 points out, the proposed method does not seem very practical for any particular application, and the authors do not give a practical demonstration that shows the usefulness of the approach. It's not clear how the paper should be positioned with respect to previous work, and the proposed method is not directly compared with standard steganography schemes on metrics such as bandwidth, robustness etc, making it difficult to assess the value of the contribution. For these reasons I recommend rejecting the paper in its current form.